# IMPROVED ROBUSTNESS TO ADVERSARIAL EXAMPLES USING LIPSCHITZ REGULARIZATION OF THE LOSS

## ABSTRACT

We augment adversarial training (AT) with *worst case adversarial training* (WCAT) which improves adversarial robustness by 11% over the current state-of-the-art result in the $\ell_2$ norm on CIFAR-10. We obtain verifiable average case and worst case robustness guarantees, based on the expected and maximum values of the norm of the gradient of the loss. We interpret adversarial training as Total Variation Regularization, which is a fundamental tool in mathematical image processing, and WCAT as Lipschitz regularization.

## 1 INTRODUCTION

We augment adversarial training (AT) with *worst case adversarial training* (WCAT) which improves adversarial robustness by 11% over the current state-of-the-art result (Qian & Wegman, 2018) in the $\ell_2$ norm. The method also achieves results comparable to the state-of-the-art results of Madry et al. (2017) in the $\ell_\infty$ norm. Moreover, our adversarial training step uses only one gradient evaluation compared to seven steps in the Madry et al. (2017) work. The worst case adversarial training method is described as follows. During adversarial training, the gradient of the loss is computed for each perturbed image. WCAT records the largest of the gradients norms, and adds a penalty to the loss proportional to this term. In many cases we observe that models trained with AT and WCAT have improved test/validation error over the unregularized model.

In §2 we show that the norm of the gradient of the loss of the model is a measure of the robustness of a model to adversarial examples. We obtain verifiable worst and average case robustness guarantees, based on the expected and maximum values of the norm of the gradient of the loss. We then compute these quantities empirically on trained models, and demonstrate that improving these quantities leads to proportional improvements in adversarial robustness.

In §3 we interpret adversarial training as Total Variation (TV) Regularization, which is a fundamental tool in mathematical image processing. TV regularization was introduced for image denoising (Rudin et al., 1992). It is a measure of the variation of a function, allowing for discontinuities. We also show that WCAT corresponds to *Lipschitz regularization*, which appears in Image Inpainting (Bertalmio et al., 2000) and function approximation (Crandall et al., 2001; Oberman, 2005). Lipschitz regularization was used in a recent proof of generalization of deep neural networks (Oberman & Calder, 2018). Write $\ell(x) = \ell \circ f(x)$ for the loss of the model. We show that training with AT and WCAT is equivalent to minimizing

$$J[\ell] = \underbrace{\mathbb{E}_{(x,y)\sim\mathcal{D}}[\ell(x)]}_{\text{expected loss}} + \epsilon \underbrace{\mathbb{E}_{(x,y)\sim\mathcal{D}}[\|\nabla_x\ell(x)\|_*]}_{\text{AT = TV regularization}} + \lambda \underbrace{\max_{(x,y)\in\mathcal{D}}\|\nabla_x\ell(x)\|_*}_{\text{WCAT = Lipschitz regularization}} \tag{1}$$

where $\epsilon$ is the size of the adversarial training perturbation, and $\lambda$ is the WCAT multiplier. The dual norm $\|\cdot\|_*$ corresponds to $\|\cdot\|_1$ for attacks measured in $\ell_\infty$ and to $\|\cdot\|_2$ for attacks measured in $\ell_2$,[1] see §3.1.

### 1.1 ADVERSARIAL ATTACKS AND ADVERSARIAL TRAINING

The earliest and most successful defense is adversarial training (Szegedy et al. (2013); Goodfellow et al. (2014); Tramèr et al. (2018); Madry et al. (2017)). Top entries in a recent adversarial defence

---

[1] We overload '$\ell$' as notation for both the loss and for norms: the meaning should be clear from the context.

competition (Kurakin et al. (2017)) used Ensemble Adversarial Training (Tramèr et al. (2018)), where a model is adversarially trained with inputs generated by an ensemble of other models. For more background, refer to the recent review Goodfellow et al. (2018) which discusses defences against adversarial attacks and their limitations.

Adversarial attacks seek to find the minimum norm vector which leads to a misclassification by the model. Finding the optimal attack is intractable (Athalye et al., 2018). An alternative which permits loss gradients to be used is to consider the attack vector measured in a given norm which most increases the loss, $\max_{\|a\| \leq \varepsilon} \ell(x + a)$.

*Adversarial training* improves robustness to adversarial attacks by solving the minimax problem

$$\min_{w} \mathbb{E}_{(x,y)\sim\mathcal{D}} \left[ \max_{\|\delta\|\leq\varepsilon} \ell(f(x + \delta; w), y) \right]$$

In practice, a tractable attack vector, $a(x)$, is used, in place of the optimal $\delta$, leading to

$$\mathbb{E}_{(x,y)\sim\mathcal{D}} \left[ \ell\left(x + a(x)\right) \right].$$

## 2  ROBUSTNESS GUARANTEES FROM THE LIPSCHITZ CONSTANT

Weng et al. (2018) and Hein & Andriushchenko (2017) showed that the Lipschitz constant of the model can be used to establish rigorous worst-case bounds on adversarial robustness. In particular Weng et al. (2018) show that the Lipschitz constant of the model gives a certifiable minimum adversarial distance: a successful attack on image $x$ will have adversarial distance at least

$$\delta \geq \min_{j \neq i^*} \frac{f_{i^*}(x) - f_j(x)}{2L_f} \tag{2}$$

where $L_f$ is the Lipschitz constant of the model, $f$, and $i^*$ is the correct label of $x$. Thus training models to have small Lipschitz constant should improve adversarial robustness (Hein & Andriushchenko (2017); Tsuzuku et al. (2018)).

*Remark* 2.1 (Application of empirical Lipschitz constants). In theory, it is possible to extend the data with a function whose Lipschitz constant matches that of the data see §C for more details. We argue that a model whose Lipschitz constant better approximates the Lipschitz constant of the data brings us closer to the ground truth. Lipschitz regularization brings the (estimated) Lipschitz constant of a model close to the Lipschitz constant of the data on which it was trained. For example on CIFAR-10, the Lipschitz constant for the dataset is 0.36. For a regularized model (ResNeXt-34, see §5) the estimated Lipschitz constant was 1.32, but was 13.70 for an undefended model (the latter two measured on the test/validation set).

### 2.1  ADVERSARIAL ROBUSTNESS BASED ON GRADIENTS OF MODEL LOSS

Here we obtain adversarial robustness bounds based on the gradient of the *loss* of the model. The second and third terms in (1) estimate the average and worst case robustness. The first part of the following result is the analogue of (2), with the model loss instead of the model (which is a scalar instead of a vector). The second part of the result gives an average case robustness bound.

**Lemma 2.2** (Worst-case and expected stability). *Write $\ell(x) = \ell \circ f(x)$ for the loss of the model. Let $a$ be any adversarial perturbation of norm $\|a(x)\| \leq \varepsilon$. If $\ell(x)$ is L-Lipschitz continuous then*

$$\ell(x + a) \leq \ell(x) + L\varepsilon \tag{3}$$

*In addition, if $TV(\ell) = \mathbb{E}_{(x,y)\sim\mathcal{D}} \|\nabla\ell(x)\|_* \leq R$ then*

$$\mathbb{E}_{(x,y)\sim\mathcal{D}}[\ell(x + a)] \leq \mathbb{E}_{(x,y)\sim\mathcal{D}}[\ell(x)] + R\varepsilon + \mathcal{O}(\varepsilon^2) \tag{4}$$

The proof is in §B.1

*Remark* 2.3 (Application of robustness guarantees). One possible use of the result is that we can estimate adversarial robustness on unseen data drawn from the same distribution using the values of the terms corresponding to AT and WCAT in (1)

For example, we estimated the values of the terms corresponding to AT and WCAT in (1) for a regularized model trained with (1) and for an undefended model. We expect the ratio of these values between the models to be an effective predictor of the relative robustness of each model. The regularized model had better values of AT and WCAT by factors of 3.3, and 9, respectively. The median adversarial distance of the regularized model was better by a factor of 5, which lies between those factors. See §5 for more empirical results.

## 3  REGULARIZATION INTERPRETATION OF AT AND WCAT

### 3.1  ADVERSARIAL TRAINING CORRESPONDS TO TV REGULARIZATION

The next result interprets adversarial training using either the one-step Signed Gradient attack vector (Goodfellow et al., 2018) or the gradient attack vector as Total Variation regularization.

**Lemma 3.1.** *Adversarial training using the $\varepsilon$-scaled one step attack vector is equivalent up to terms of order $\varepsilon^2$ to augmenting the loss with Total Variation regularization,*

$$J[\ell] = \mathop{\mathbb{E}}_{(x,y)\sim\mathcal{D}} [\ell(x) + \varepsilon\|\nabla\ell(x)\|_*]$$

*where the $\|\cdot\|_*$ is the dual norm to the norm measuring adversarial perturbations.*

The proof is in §B.2.

### 3.2  WORST CASE AT CORRESPONDS TO LIPSCHITZ REGULARIZATION

The basis for this result is Rademacher's Theorem (Evans, 2018, §3.1), which states that if a function $g(x)$ is Lipschitz continuous then it is differentiable almost everywhere and

$$\mathrm{Lip}(g) = \max_x \|\nabla g(x)\|$$

We obtain an underestimate of $\mathrm{Lip}(g)$ by sampling the norm of the gradient on a subset of points.

**Definition 3.2** (Method for estimating the Lipschitz constant). Let $\ell$ be a Lipschitz continuous function. Then

$$\max_{x\in\mathcal{D}} \|\nabla\ell(x)\| \leq \mathrm{Lip}(\ell) \tag{5}$$

During training, we apply (5) to $\ell(x)$ and set $\mathcal{D}$ to be a mini-batch to obtain the WCAT term in (1). When we estimate the Lipschitz constant of a trained model, we use the full test/validation dataset.

*Remark* 3.3. Regularization of the loss corresponds to partially regularizing the model, but at a much lower cost. Since the loss is a scalar, regularizing by the Lipschitz constant of the loss is equivalent to regularization of the model $f$ in one direction. By the chain rule,

$$\nabla_x \ell(f(x), y) = \nabla_f \ell(f(x), y) \, \nabla_x f(x)$$

For example, when $\ell$ is the KL divergence, and when $f = \mathrm{softmax}(z(x))$ then

$$\nabla_x \ell(f(x), y) = (f(x) - y) \, \nabla_x z(x)$$

Thus, in this case, regularizing $\ell(x)$ corresponds to regularization of $z(x)$ in the direction $f(x) - y$.

## 4  IMPLEMENTATION OF THE LIPSCHITZ CONSTANT OF A NETWORK

Data independent upper bounds on the Lipschitz constant of the model go back to Bartlett (1996). These bounds are based on the product of the norm of the weight matrices, and neglect the effects of the activation function. We summarize this upper bound as follows. Let $W^k$ be the weight matrix of the $k$-th layer of a network $f$ comprised of $N$ layers, and suppose all non linearities of a network are at most 1-Lipschitz. Then

$$\mathrm{Lip}_{p,q}(f) \leq \prod_{k=1}^{N} \|W^k\|_{p_k, p_{k-1}} \tag{6}$$

with $p_0 = p$ and $p_n = q$. Certain conditions on the $p_k$'s must be met. For a proof with $p, q = 2$ see Tsuzuku et al. (2018).

Recent works based on this estimate include: Cissé et al. (2017), using the distance to the nearest orthonormal matrix; Miyato et al. (2018), using the 2-norm; Gouk et al. (2018), using either the 1-norm or $\infty$-norm; and Qian & Wegman (2018), using the $\infty$-norm. In these works, the estimate is used to penalize the Lipschitz constant of the model during training, and so the the logarithm of the estimate is used. Of these four papers, only the implementation of Gouk et al. accounts for batch normalization. Since batch normalization multiplies each weight matrix by a diagonal scaling matrix, the other works are missing important terms in their implementations.

A different method for estimating the Lipschitz constant of a pre-trained model was presented in Weng et al. (2018), using statistical techniques from extreme value theory. This estimate accurately captures the minimum adversarial distance needed to successfully misclassify an image. However it requires at a minimum many tens of model evaluations for each image, and so is not tractable as a Lipschitz penalty during training.

### 4.1 COMPARING METHODS

Interpreting (5) as Lipschitz regularization leads to a more accurate and efficient method for estimating the Lipschitz constant in a deep neural network, compared to other recent methods based on the product of weight matrix norms, whose error grows exponentially in the number of layers. For deep models, (6) has a large gap: on the models we considered, this estimate was no smaller than $10^{12}$ and as large as $10^{23}$. In contrast, our empirical results, which are a lower bound (see §D), give values less than 10. The fact that the robustness guarantees of §2 using (5) give meaningful results suggests that (5) is an accurate estimate for our purposes.

## 5 EMPIRICAL RESULTS

We studied image classification on the CIFAR-10 and CIFAR-100 datasets (Krizhevsky & Hinton (2009)). We tested our methods on three networks, chosen to represent a broad range of architectures: AllCNN-C (Springenberg et al. (2014)), having nine layers; a 34 layer ResNet (He et al. (2016)); and a 34 layer ResNeXt (Xie et al. (2017)). Training and model details are provided in Appendix A.

In §5.1 we define error curves, an error metric which allows for easy comparisons of model robustness and attack strength across a full range of attack norms. Then in §5.2 we use these curves to rank common attack methods. The curves illustrate a clear ranking of attack methods across models. The change in attack ranking is based on the norm used: Iterative FGSM ($\ell_\infty$ PGD) is the most effective attack when distance is measured in the $\ell_\infty$ norm; $\ell_2$ PGD is the most effective attack when distance is measured in the $\ell_2$ norm. The attack ranking makes defence robustness easier to assess, since we only need to check one attack per norm for each model. In §5.3 we compare the different choices of adversarial defences discussed in this paper, which amounts to a combination of adversarial training and Lipschitz regularization in the corresponding norm. Finally, in §5.4 we compare our results to other works.

### 5.1 ERROR CURVES FOR MODELS AND ROBUSTNESS METRICS

The error curve of a model for a given attack measures the robustness of the model as a function of the norm of the attack.

**Definition 5.1.** The error curve $C_{\text{err}}(\varepsilon)$ of the model $f$ for the attack $a$ is the probability over the test data $\mathcal{D}$ that an attack of size $\varepsilon$ leads to a misclassification

$$C_{\text{err}}(\varepsilon) = P_{\mathcal{D}}\{c(x + a(x)) \neq c^*(x) \mid \text{ for an attack } \|a(x)\|_X \leq \varepsilon\}$$

where $c(x)$ is the model's label at $x$ and $c^*(x)$ is the true label.

See Figure 1 for error curves for an undefended model over a variety of attacks.

We can compare how models or attacks perform using the model error curve. We report common adversarial statistics, which can be read off the curve, in Table 1. For example the percent misclas-

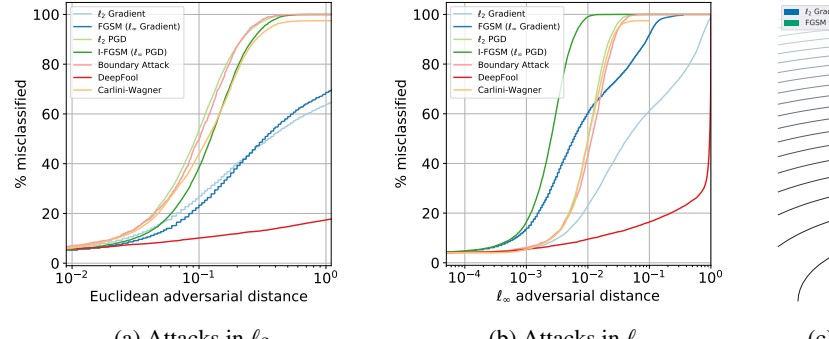

(a) Attacks in $\ell_2$      (b) Attacks in $\ell_\infty$      (c) Gradient directions

Figure 1: (1a) and (1b): Comparison of attack methods using error curves for undefended ResNeXt-34, on the CIFAR-10 test set. A higher curve means more probability of error. The $\ell_2$ projected gradient method is the most effective attack when measured in $\ell_2$; iterative FGSM ($\ell_\infty$ PGD) is the most effective attack when measured in $\ell_\infty$. (1c): Comparison of Iterative FGSM and $\ell_2$ gradient ascent on a quadratic function in two dimensions.

sification at adversarial distance $\varepsilon = 0$ and $0.1$ (in the 2-norm) are easily read off the error curve. These values of $\varepsilon$ correspond to the test error, and noise which is slightly smaller than a human perceptible perturbation. We also report the median $\ell_2$ distance which corresponds to the $x$-intercept of 50% error on the curve.

## 5.2 ATTACK EVALUATION

We attacked models on the test/validation set using seven untargeted attack methods: gradient attack; projected gradient descent (constrained in $\ell_2$); the Fast Gradient Sign Method (FGSM) (Goodfellow et al. (2014)); Iterative FGSM (I-FGSM) (Kurakin et al. (2016)), or projected gradient descent in $\ell_\infty$; DeepFool (Moosavi-Dezfooli et al. (2016)); the Carlini-Wagner attack (Carlini & Wagner, 2017); and the Boundary attack (Brendel et al., 2018). The first six methods are first order gradient based white-box attacks, while the last is a black-box attack. I-FGSM and the $\ell_2$ projected gradient attack are iterative gradient methods, whereas FGSM and the gradient attack are single step. All attacks were implemented with Foolbox (Rauber et al. (2017)). Hyperparameters were set to Foolbox defaults, except for the Boundary attack[2]. For each image and attack method, we calculated the adversarial distance in $\ell_2$ and $\ell_\infty$.

On each model, dataset, and regularization method, we tested all seven attack methods on the entire test/validation set. We compared attack methods using the attack error curve. For example see Figure 1, where we plot attack error curves for each attack method on an undefended model. We plot the attack error curves in both Euclidean and $\ell_\infty$ distances.

When attack norms are measured in $\ell_\infty$, the most effective attack is Iterative FGSM (PGD in $\ell_\infty$). Similarly, when attack norms are measured in $\ell_2$, $\ell_2$ projected gradient descent is the strongest attack. See Figure 1 for an illustration. We observed the same ranking of attacks on all models and defences studied. For this reason in what follows, we only report model statistics using projected gradient descent corresponding to the distance metric used (either $\ell_2$ or $\ell_\infty$).

## 5.3 EVALUATION OF DEFENCE METHODS

We tested many combinations of defence methods. We use the superscipts $0, 1, 2$ to indicate the type of adversarial training used: 0 for none, 1 for one step FGSM, and 2 for one step gradient ascent. We used the superscript Lip for Lipschitz regularization, which measured the gradient in the same norm as was used for adversarial training. For example $J^{2-\text{Lip}}$ indicates adversarial training and Lipschitz regularization in the $\ell_2$ norm.

---

[2] The Boundary attack is a computationally demanding attack. Instead of attacking all test images, we only attacked the first 1000 images, for 5000 iterations.

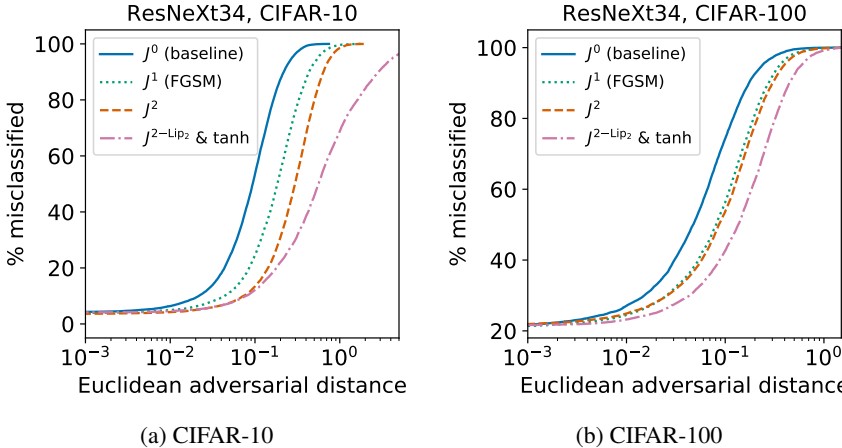

(a) CIFAR-10           (b) CIFAR-100

Figure 2: Adversarial robustness results against $\ell_2$ PGD attacks using ResNeXt networks on CIFAR 10 and 100. The curves are for the undefended (baseline) model, and different regularizations, showing the error rates at different attack vectors norms. The $\ell_2$ PGD attack was the most effective in the $\ell_2$ norm.

We also considered adding a final sigmoid layer to the network, prior to the $\mathrm{softmax}$. The choice of sigmoid we choose is $\tanh$, and is inspired by (but not equivalent to) $\tanh$-estimators used in classical statistics as a robust estimator (Hampel et al., 2011, Chapter 2). The intuition behind this choice is to normalize the logit scores of the model, which we believe should improve robustness to outliers. Outside of deep learning, $\tanh$-estimators have been successfully used to normalize scores and improve robustness, for example in machine learning biometrics (Jain et al. (2005)). See Appendix A.1 for layer details.

Model robustness is evaluated on the entire test/validation set using the median adversarial distance in both $\ell_2$ and $\ell_\infty$, and the percent misclassified at adversarial $\ell_2$ distance $\varepsilon = 0.1$. We chose $\varepsilon = 0.1$ because at this magnitude attacks are still imperceptible to the human eye. For ease of comparison with other works, we also report percent misclassified at $\ell_2$ distance $\varepsilon = 1.5$, and (on CIFAR-10) $\ell_\infty$ distance $\varepsilon = \frac{8}{255}$. In addition we plot the attack error curve for each model. Table 1 and Figure 2 present a summary of results for ResNeXt-34. The best statistics are in bold. Complete results, for all models, datasets, and adversarial defences, are presented in Appendix D.

Here we summarize our results for ResNeXt-34, the model studied with the greatest capacity using the most effect attack in the $\ell_2$ norm, $\ell_2$ PGD. Refer also to Table 1 for a summary. Without adversarial perturbations, all ResNeXt-34 models achieve roughly 4% test error on CIFAR-10. However, the undefended (baseline, $J^0$) model achieves 54% test error at adversarial $\ell_2$ distance $\varepsilon = 0.1$. Adversarial training via FGSM ($J^1$) reduces test error to 24.6%, whereas $\ell_2$ adversarial training ($J^2$) reduces test error to 13.5%. A combination of all defences ($J^{2-\mathrm{Lip}}$ with $\tanh$) further reduces test error to 12.1%. The models are ranked in the same order when instead measured with median $\ell_2$ adversarial distance. The model with all defences has median adversarial distance six times that of the undefended model. FGSM ($J^1$) only doubles the median adversarial distance relative to the baseline undefended model. Figure 2a illustrates that this ranking of defences holds over all distances of adversarial perturbations.

We observe a similar ranking on CIFAR-100. See for Figure 2b. Unperturbed, all models achieve between 21% and 22% test error. Without adversarial defences, ResNeXt-34 (4x32d) has a test error of 74% at adversarial $\ell_2$ distance $\varepsilon = 0.1$. Adversarial training alone brings the test error down to 56.3% and 53.7%, with respectively FGSM ($J^1$) and $\ell_2$ ($J^2$) adversarial training. A combination of all defences further reduces test error to 42.6%. Median $\ell_2$ adversarial distance increased from 0.05 on the undefended model to 0.14 on the model with all defences.

The test/validation error of the regularized models is in many cases better than the baseline undefended model. On CIFAR-10 AT and WCAT can improve test/validation error by nearly one percent

Table 1: Adversarial statistics with ResNeXt-34. The columns $\|\nabla \ell\|_2$ and $\|\nabla f\|_{2,\infty}$ report the maximum observed norm on the test data.

| Dataset | defense method | Euclidean distance | | max test statistics | |
|---|---|---|---|---|---|
| | | median distance | % Err at $\varepsilon = 0.1$ | $\|\nabla \ell\|_2$ | $\|\nabla f\|_{2,\infty}$ |
| CIFAR-10 | $J^0$ (baseline) | 0.09 | 53.98 | 85.21 | 13.70 |
| | $J^1$ (AT, FGSM) | 0.18 | 24.63 | 35.77 | 6.27 |
| | $J^2$ (AT, $\ell_2$) | 0.30 | 13.54 | 32.13 | 5.22 |
| | $J^{2-\text{Lip}}$ & tanh | **0.56** | **12.12** | 9.22 | 1.32 |
| CIFAR-100 | $J^0$ (baseline) | 4.74e$-$2 | 74.18 | 93.83 | 1.89 |
| | $J^1$ (AT, FGSM) | 8.08e$-$2 | 56.34 | 34.60 | 0.71 |
| | $J^2$ (AT, $\ell_2$) | 8.61e$-$2 | 53.77 | 44.81 | 0.73 |
| | $J^{2-\text{Lip}}$ & tanh | **0.136** | **42.58** | 17.97 | 0.35 |

(see AllCNN results in Table 3). On CIFAR-100 some regularized models perform slightly worse, but typically the difference is no more than one or two percent.

In Table 1 we also report statistics measuring the model's Lipschitz constant. For vector valued functions, $f$, in order to measure the Lipschitz constant the induced matrix norm[3] is used, coming from the norms in the source and target spaces. Further statistics for all adversarial defences and models are deferred to Appendix D. The columns $\|\nabla l\|_2$ and $\|\nabla f\|_{2,\infty}$ give the maximum of these norms over the test/validation set. Employing all defences significantly decreases the norm of the model Jacobian on the test data, and hence improves model robustness. On CIFAR-10 the model with all defences has Jacobian norm 10 times smaller than the undefended model, whereas adversarial training only improves the Jacobian norm by a factor of three at most. Similarly on CIFAR-100, adversarial training alone improves the norm of the Jacobian by a factor of no more than three. However a combination of all defences decreases the norm of the model Jacobian by a factor of six. These statistics indicate that 2-Lipschitz regularization of the loss, combined with $\ell_2$ adversarial training, dramatically reduce the Lipschitz constant of a network measured in the $2, \infty$ norm.

Interestingly, 2-Lipschitz regularization of the loss also decreases the $1, 1$ Jacobian norm of the model (see Tables 4 and 6). Thus 2-Lipschitz regularization improves robustness to $\ell_\infty$ attacks as well.

In Appendix D we report results for all models and combinations of defence methods. Of the individual defences by themselves, adversarial training ($J^1$ or $J^2$) improves model robustness the most. We find $\ell_2$ adversarial training ($J^2$) to be more effective than FGSM ($J^1$) when attacks are measured in $\ell_2$. We observe the same ranking of defence methods for AllCNN and ResNet-34. Adversarial training improves model robustness. However model robustness is further improved by adding Lipschitz regularization of the loss, which empirically decreases the Jacobian norm of the model on the test data.

In terms of training time, both adversarial training and Lipschitz regularization increase training time by a factor of no more than four. In contrast, adding a final tanh layer to normalize the logits is nearly free, and consistently improves model robustness by itself.

## 5.4 COMPARISON WITH OTHER WORK

On the CIFAR-10 dataset (as of revision) the current state-of-the-art $\ell_2$ robustness is Qian & Wegman (2018). Without adversarial training, their method achieves 89.9% error at $\varepsilon = 1.5$. In contrast, without adversarial training, Lipschitz regularization as implemented here achieves 78.42% error at $\varepsilon = 1.5$, an improvement of over 11%. With adversarial training, our results are comparable to Qian & Wegman (2018): they achieved 79.6% error at adversarial perturbation $\varepsilon = 1.5$, whereas our method achieves 78.59%, an improvement of 1%. A direct comparison is difficult, since Qian & Wegman used the Carlini-Wagner attack, which (with Foolbox defaults) is a weaker $\ell_2$ attack

---

[3]The induced norm of a matrix $A$ is defined as $\|A\|_{p,q} = \max_{x \neq 0} \frac{\|Ax\|_q}{\|x\|_p}$ (Horn et al., 1990, Chapter 5.6.4)

than the attack used here ($\ell_2$ PGD), so our results should be adjusted accordingly. (We applied the Carlini-Wagner attack but we only recorded the results for the strongest attack).

After revision we ran comparisons for attacks measured in $\ell_\infty$. We only used one step adversarial training. The current state-of-the-art on CIFAR-10 is Madry et al. (2017) which used seven steps of $\ell_\infty$ PGD. The strongest attack achieves 54.2% error at $\varepsilon = \frac{8}{255}$. Our best model, trained with $J^{2-\mathrm{Lip}}$ regularization, is slightly worse, with 62.4% error at $\varepsilon = \frac{8}{255}$. We expect that multi-step adversarial training would improve our results in this setting.

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

## A  MODEL AND TRAINING DETAILS

We used standard data augmentation for the CIFAR dataset, comprising of horizontal flips, and random crops of padded images, four pixels per side. Images are scaled to have pixel values in $[0, 1]$. We used square cutout (Devries & Taylor (2017)) of width 16 on CIFAR-10, and width 8 on CIFAR-100, but no dropout. Batch normalization was used after every convolution layer. We used SGD with an initial learning rate of 0.1, momentum set to 0.9, and a batch size of 128. CIFAR-10 was trained for 200 epochs, dropping the learning rate by a factor of five after epochs 60, 120, and 180. On CIFAR-100, networks were trained for 300 epochs, and the learning rate was dropped by a factor of 10 after epochs 150 and 225. For CIFAR-10 weight decay (Tikhonov/$\ell_2$ regularization) was set to $5e{-}4$; on CIFAR-100 it was $1e{-}4$.

For networks with Lipschitz regularization in the 2-norm, the Lagrange multiplier $\lambda$ of the excess Lipschitz term was set to $\lambda = 0.1$. With 1-norm Lipschitz regularization, the Lagrange multiplier was scaled down by a factor of $\sqrt{n}$ from the 2-Lipschitz Lagrange multiplier, where $n$ is the input dimension. Adversarially trained models were trained with images perturbed to an $\ell_2$ distance of $\varepsilon = 0.01$. We did not tune either of these hyperparameters.

For CIFAR-10, the ResNeXt architecture we used had a depth of 34 layers, cardinality 2 and width 32, with a basic residual block rather than a bottleneck. The branches (convolution groups) of the blocks were aggregated via a mean, rather than using a fully connected layer. For CIFAR-100 the architecture was the same, but had cardinality 4.

### A.1  PRE-softmax SIGMOID LAYER

Prior to the final $\mathrm{softmax}$ layer, we found inserting a sigmoid activation function improved model robustness. In this case, the sigmoid layer comprised of first batch normalization (without learnable parameters), followed by the activation function $t\tanh(\frac{x}{t})$, where $t$ is a single learnable parameter, common across all layer inputs.

Other possible defences discussed in Goodfellow et al. (2018) include input validation and preprocessing, which would potentially allow adversarial samples to be recognized before being input to the model, and architecture modifications designed to improve robustness to adversarial samples. For more information we refer to the review (Goodfellow et al. (2018)) and the discussion of attack methods in (Brendel et al. (2018)).

## B  PROOFS

### B.1  PROOF OF LEMMA 2.2

*Proof.* By Lipschitz continuity of $\ell$
$$|\ell(x + a) - \ell(x)| \leq L\|a\| = L\varepsilon$$
There are two cases for the left-hand side, depending on the sign. In both cases we obtain (3).

Using Taylor's theorem
$$
\begin{aligned}
\ell(x + a(x)) &= \ell(x) + \nabla\ell(x) \cdot a(x) + \mathcal{O}(\varepsilon^2) \\
&\leq \ell(x) + \|\nabla\ell(x)\|_* \|a(x)\| + \mathcal{O}(\varepsilon^2) \qquad \text{by dual norm §3.1} \\
&\leq \ell(x) + \varepsilon \|\nabla\ell(x)\|_* + \mathcal{O}(\varepsilon^2)
\end{aligned}
$$
Taking expectations, we obtain (4). □

Table 2: Lipschitz constants of common training sets. CIFAR-100 has several duplicated images with different labels, these were removed from the calculation.

| Dataset | MNIST | FashionMNIST | CIFAR-10 | CIFAR-100 |
|---|---|---|---|---|
| $\mathrm{Lip}_{2,\infty}(\mathcal{D})$ | 0.417 | 0.626 | 0.364 | 1.245 |

## B.2 PROOF OF LEMMA 3.1

The following result shows that Signed Gradient (Goodfellow et al., 2014) attack, and the normalized gradient attack are nearly optimal in the following sense.

**Lemma B.1.** *Write $\ell(x) = \ell(f(x), y)$. The attack vector directions*

$$(a)_i = \frac{\nabla\ell(x)_i}{|\nabla\ell(x)_i|}, \qquad and \qquad a = \frac{\nabla\ell(x)}{\|\nabla\ell(x)\|_2},$$

*are nearly optimal the $\ell_\infty$ and $\ell_2$ norm, respectively, in the sense that*

$$\ell(x + \varepsilon a) = \max_{\|b\|_p \leq \varepsilon} \ell(x + b) + \mathcal{O}(\varepsilon^2), \qquad p = \infty, 2, \text{ respectively}$$

*and moreover,*

$$\ell(x + \varepsilon a) = \ell(x) + \varepsilon\|\nabla\ell(x)\|_p + \mathcal{O}(\varepsilon^2), \qquad p = 1, 2, \text{ respectively} \tag{7}$$

First observe that the proof of Lemma 3.1 follows by taking expectations in (7). Next we prove Lemma B.1.

The optimal attack solves

$$\max_{\|a\| \leq \varepsilon} \ell(x + a) \tag{8}$$

*Proof.* Use the Taylor expansion

$$\ell(x + \varepsilon a) = \ell(x) + \varepsilon a \cdot \nabla_x \ell(x) + \mathcal{O}(\varepsilon^2)$$

Thus, the optimal attack vector defined by (8) in a generic norm $\|\cdot\|$ can be approximated to order $\varepsilon^2$ by solving the problem

$$\max_{\|a\| \leq 1} \nabla_x \ell(f(x), y) \cdot a$$

According to (Boyd & Vandenberghe, 2004, A.1.6), the equation

$$\max_{\|a\| \leq 1} b \cdot a = \|b\|_*$$

where the norm on the right hand size is the dual norm. Thus

$$\max_{\|a\| \leq \varepsilon} \nabla_x \ell(f(x), y) \cdot a = \varepsilon\|\nabla_x \ell(f(x), y)\|_p, \qquad p = 1, 2 \text{ respectively.} \qquad \square$$

## C LIPSCHITZ CONSTANT OF DATA AND OPTIMAL EXTENSIONS

Define the Lipschitz constant of the data (in the $2, \infty$ norm) to be

$$\mathrm{Lip}_{2,\infty}(\mathcal{D}) = \max_{x_1, x_2 \in \mathcal{D}} \left\{ \frac{\|c^*(x_1) - c^*(x_2)\|_\infty}{\|x_1 - x_2\|_2} \;\middle|\; c^*(x_1) \neq c^*(x_2) \right\}$$

Table 2 lists the Lipschitz constant of the training data for common datasets, which are all small: all but one are below 1 in the $2, \infty$ norm.

The Lipschitz extension theorem (Valentine, 1945) says that given function values $\{f(x)\}_{x \in \mathcal{D}}$, there exists an extension $f_{\mathrm{ext}}$ which perfectly fits the data, and has the same Lipschitz constant, provided the appropriate norm are used on the $X$ and $Y$ spaces. This can be done using, for example, the 2-norm for $X$ and the $\infty$ norm on the label space. In other norms, we can also make an extension, but the Lipschitz constant may increase (Johnson & Lindenstrauss, 1984). Of course, such a function may not be consistent with a given architecture.

See Table 2, where we present the Lipschitz constant of common datasets.

Table 3: CIFAR-10 adversarial statistics

| Model | defense method | % Err at $\varepsilon = 0$ | Euclidean adversarial distance | | | $\ell_\infty$ adversarial distance | |
|---|---|---|---|---|---|---|---|
| | | | median distance | % Err at $\varepsilon = 0.1$ | % Err at $\varepsilon = 1.5$ | median distance | % Err at $\varepsilon = \frac{8}{255}$ |
| AllCNN | $J^0$ | 6.01 | 0.13 | 38.11 | 94.10 | 4.3e−3 | 98.89 |
| | $J^{0-\mathrm{Lip}_1}$ | 6.37 | 0.22 | 23.5 | 94.32 | 6.3e−3 | 89.83 |
| | $J^{0-\mathrm{Lip}_2}$ | 6.26 | 0.17 | 29.27 | 94.02 | 5.3e−3 | 91.19 |
| | $J^0$ & tanh | 5.41 | 0.19 | 32.61 | 89.51 | 6.8e−3 | 80.91 |
| | $J^{0-\mathrm{Lip}_1}$ & tanh | 5.69 | 0.24 | 21.85 | 95.63 | 7.1e−3 | 84.71 |
| | $J^{0-\mathrm{Lip}_2}$ & tanh | 5.45 | 0.21 | 25.04 | 90.8 | 6.8e−3 | 84.45 |
| | $J^1$ | 5.30 | 0.21 | 24.4 | 92.4 | 6.5e−3 | 87.40 |
| | $J^{1-\mathrm{Lip}_1}$ | 6.08 | 0.24 | 21.61 | 92.99 | 7.1e−3 | 88.13 |
| | $J^1$ & tanh | **5.05** | 0.26 | 22.25 | 95.91 | 8.8e−3 | 75.78 |
| | $J^{1-\mathrm{Lip}_1}$ & tanh | 5.47 | 0.29 | 18.85 | 92.52 | 8.8e−3 | 80.45 |
| | $J^2$ | 5.90 | 0.29 | 17.09 | 86.38 | 9.4e−3 | 81.13 |
| | $J^{2-\mathrm{Lip}_2}$ | 5.84 | 0.29 | 16.86 | 88.63 | 9.2e−3 | 83.10 |
| | $J^2$ & tanh | 5.10 | **0.38** | 16.19 | **81.58** | **13.4e−3** | **68.97** |
| | $J^{2-\mathrm{Lip}_2}$ & tanh | 5.27 | 0.35 | **15.00** | 83.46 | 11.8e−3 | 73.73 |
| ResNet34 | $J^0$ | 6.00 | 0.09 | 56.00 | 100 | 2.3e−3 | 100 |
| | $J^{0-\mathrm{Lip}_1}$ | 6.81 | 0.19 | 24.75 | 100 | 5.2e−3 | 100 |
| | $J^{0-\mathrm{Lip}_2}$ | **5.43** | 0.17 | 27.08 | 100 | 4.8e−3 | 100 |
| | $J^0$ & tanh | 5.54 | 0.20 | 34.44 | 97.41 | 7.7e−3 | 89.90 |
| | $J^{0-\mathrm{Lip}_1}$ & tanh | 7.3 | 0.25 | 25.56 | 94.82 | 7.3e−3 | 88.67 |
| | $J^{0-\mathrm{Lip}_2}$ & tanh | 6.14 | 0.21 | 28.66 | 94.47 | 6.7e−3 | 88.59 |
| | $J^1$ | 5.68 | 0.16 | 30.31 | 100 | 4.4e−3 | 97.87 |
| | $J^{1-\mathrm{Lip}_1}$ | 7.02 | 0.22 | 22.46 | 100 | 5.9e−3 | 95.96 |
| | $J^1$ & tanh | 5.94 | 0.24 | 25.96 | 99.51 | 8.4e−3 | 92.33 |
| | $J^{1-\mathrm{Lip}_1}$ & tanh | 6.87 | 0.28 | 21.47 | 96.21 | 8.4e−3 | 88.94 |
| | $J^2$ | 5.57 | 0.25 | 18.19 | 99.95 | 7.4e−3 | 99.09 |
| | $J^{2-\mathrm{Lip}_2}$ | 5.65 | 0.28 | 16.74 | 99.99 | 7.9e−3 | 87.72 |
| | $J^2$ & tanh | 5.52 | **0.46** | 17.45 | 93.2 | **17.1e−3** | **71.55** |
| | $J^{2-\mathrm{Lip}_2}$ & tanh | 5.81 | 0.40 | **15.84** | **91.27** | 14.3e−3 | 77.71 |
| ResNeXt34 (2x32d) | $J^0$ | 4.07 | 0.09 | 53.98 | 100 | 2.7e−3 | 100 |
| | $J^{0-\mathrm{Lip}_1}$ | 5.36 | 0.22 | 20.35 | 100 | 5.9e−3 | 99.97 |
| | $J^{0-\mathrm{Lip}_2}$ | 4.28 | 0.21 | 19.13 | 100 | 5.8e−3 | 99.96 |
| | $J^0$ & tanh | 4.05 | 0.34 | 23.97 | 87.75 | 13.4e−3 | 73.33 |
| | $J^{0-\mathrm{Lip}_1}$ & tanh | 4.70 | 0.33 | 18.7 | 86.46 | 10.5e−3 | 78.39 |
| | $J^{0-\mathrm{Lip}_2}$ & tanh | 4.18 | 0.33 | 19.64 | **78.42** | 11.1e−3 | 72.46 |
| | $J^1$ | 3.87 | 0.19 | 23.26 | 100 | 5.6e−3 | 92.74 |
| | $J^{1-\mathrm{Lip}_1}$ | 5.02 | 0.25 | 17.16 | 100 | 6.93e−3 | 91.83 |
| | $J^1$ & tanh | 4.16 | 0.36 | 20.44 | 92.87 | 12.8e−3 | 75.60 |
| | $J^{1-\mathrm{Lip}_1}$ & tanh | 4.84 | 0.34 | 16.92 | 89.23 | 10.8e−3 | 81.88 |
| | $J^2$ | **3.58** | 0.30 | 13.54 | 99.92 | 9.0e−3 | 98.34 |
| | $J^{2-\mathrm{Lip}_2}$ | 4.13 | 0.31 | 12.52 | 99.89 | 9.1e−3 | 98.10 |
| | $J^2$ & tanh | 3.80 | **0.61** | 12.71 | 87.19 | **23.2e−3** | **62.44** |
| | $J^{2-\mathrm{Lip}_2}$ & tanh | 4.08 | 0.56 | **12.12** | 78.59 | 20.4e−3 | 63.82 |

## D  FURTHER EXPERIMENTAL RESULTS

Here we present complete results for all regularization types studied, on all models and datasets considered. Adversarial distances reported in $\ell_2$ were generated using $\ell_2$ PGD; distances in $\ell_\infty$ were generated using $\ell_\infty$ PGD (Iterative FGSM).

Table 4: CIFAR-10 stability statistics. Each column reports the maximum observed norm over the test data.

| Model | defense method | $\|\nabla \ell\|_2$ | $\|\nabla f\|_{2,\infty}$ | $\|\nabla \ell\|_1$ | $\|\nabla f\|_{1,1}$ |
|---|---|---|---|---|---|
| AllCNN | $J^0$ | 13.88 | 2.12 | 490.38 | 0.71 |
| | $J^{0-\mathrm{Lip}_1}$ | 8.41 | 1.07 | 249.00 | 0.66 |
| | $J^{0-\mathrm{Lip}_2}$ | 8.71 | 1.29 | 308.19 | 0.53 |
| | $J^0$ & tanh | 13.94 | 1.70 | 460.83 | 0.54 |
| | $J^{0-\mathrm{Lip}_1}$ & tanh | 9.45 | 1.44 | 345.78 | 0.60 |
| | $J^{0-\mathrm{Lip}_2}$ & tanh | 6.88 | 0.88 | 225.87 | 0.32 |
| | $J^1$ | 12.84 | 1.70 | 374.02 | 0.65 |
| | $J^{1-\mathrm{Lip}_1}$ | 7.62 | 1.03 | 250.18 | 0.57 |
| | $J^1$ & tanh | 13.88 | 1.95 | 409.48 | 1.16 |
| | $J^{1-\mathrm{Lip}_1}$ & tanh | 7.93 | 1.15 | 280.40 | 0.78 |
| | $J^2$ | 6.15 | 0.85 | 214.54 | 0.57 |
| | $J^{2-\mathrm{Lip}_2}$ | 5.11 | 0.79 | 186.21 | 0.43 |
| | $J^2$ & tanh | 9.08 | 1.30 | 322.52 | 0.74 |
| | $J^{2-\mathrm{Lip}_2}$ & tanh | 7.94 | 1.16 | 256.49 | 0.68 |
| ResNet34 | $J^0$ | 73.73 | 11.67 | 2702.20 | 3.97 |
| | $J^{0-\mathrm{Lip}_1}$ | 15.19 | 2.58 | 604.91 | 0.78 |
| | $J^{0-\mathrm{Lip}_2}$ | 20.56 | 3.12 | 772.40 | 0.94 |
| | $J^0$ & tanh | 36.88 | 4.75 | 1307.94 | 1.41 |
| | $J^{0-\mathrm{Lip}_1}$ & tanh | 9.29 | 1.29 | 341.03 | 0.47 |
| | $J^{0-\mathrm{Lip}_2}$ & tanh | 7.69 | 0.99 | 294.28 | 0.33 |
| | $J^1$ | 32.55 | 4.87 | 1203.04 | 1.76 |
| | $J^{1-\mathrm{Lip}_1}$ | 12.31 | 1.93 | 451.99 | 0.83 |
| | $J^1$ & tanh | 22.71 | 3.16 | 897.02 | 1.26 |
| | $J^{1-\mathrm{Lip}_1}$ & tanh | 8.77 | 1.18 | 297.64 | 0.68 |
| | $J^2$ | 18.25 | 2.91 | 703.43 | 0.95 |
| | $J^{2-\mathrm{Lip}_2}$ | 12.61 | 1.88 | 476.62 | 0.76 |
| | $J^2$ & tanh | 14.98 | 2.06 | 546.27 | 0.70 |
| | $J^{2-\mathrm{Lip}_2}$ & tanh | 7.40 | 1.25 | 260.89 | 0.51 |
| ResNeXt34 (2x32d) | $J^0$ | 85.21 | 13.70 | 2729.44 | 4.06 |
| | $J^{0-\mathrm{Lip}_1}$ | 15.32 | 2.30 | 533.59 | 1.22 |
| | $J^{0-\mathrm{Lip}_2}$ | 21.39 | 3.22 | 829.35 | 1.90 |
| | $J^0$ & tanh | 43.65 | 5.87 | 1598.69 | 2.32 |
| | $J^{0-\mathrm{Lip}_1}$ & tanh | 10.09 | 1.49 | 339.86 | 0.89 |
| | $J^{0-\mathrm{Lip}_2}$ & tanh | 8.32 | 0.96 | 322.39 | 0.46 |
| | $J^1$ | 35.77 | 6.27 | 1313.00 | 1.88 |
| | $J^{1-\mathrm{Lip}_1}$ | 14.26 | 2.16 | 469.37 | 1.37 |
| | $J^1$ & tanh | 24.98 | 3.74 | 840.59 | 1.46 |
| | $J^{1-\mathrm{Lip}_1}$ & tanh | 9.14 | 1.32 | 308.25 | 0.63 |
| | $J^2$ | 32.13 | 5.22 | 1064.98 | 2.11 |
| | $J^{2-\mathrm{Lip}_2}$ | 14.10 | 2.14 | 483.84 | 1.26 |
| | $J^2$ & tanh | 18.90 | 2.40 | 663.96 | 0.89 |
| | $J^{2-\mathrm{Lip}_2}$ & tanh | 9.22 | 1.32 | 341.11 | 0.59 |

Table 5: CIFAR-100 adversarial statistics

| Model | defense method | % Err at $\varepsilon = 0$ | Euclidean distance | | $\ell_\infty$ distance | |
|---|---|---|---|---|---|---|
| | | | median distance | % Err at $\varepsilon = 0.1$ | median distance | % Err at $\varepsilon = \frac{1}{255}$ |
| AllCNN | $J^0$ | **25.25** | 6.2e−2 | 63.58 | 1.8e−3 | 72.76 |
| | $J^{0-\text{Lip}_1}$ | 26.31 | 8.9e−2 | 53.27 | 2.4e−3 | 63.31 |
| | $J^{0-\text{Lip}_2}$ | 25.89 | 7.8e−2 | 56.45 | 2.2e−3 | 66.29 |
| | $J^0$ & tanh | 26.06 | 6.1e−2 | 64.77 | 1.9e−3 | 72.23 |
| | $J^{0-\text{Lip}_1}$ & tanh | 26.4 | 8.8e−2 | 54.9 | 2.2e−3 | 65.16 |
| | $J^{0-\text{Lip}_2}$ & tanh | 26.23 | 7.9e−2 | 56.26 | 2.3e−3 | 63.83 |
| | $J^1$ | 25.81 | 8.9e−2 | 56.26 | 2.4e−3 | 63.76 |
| | $J^{1-\text{Lip}_1}$ | 25.81 | **10.7e−2** | **49.65** | **2.8e−3** | **58.99** |
| | $J^1$ & tanh | 25.81 | 8.8e−2 | 55.03 | 2.2e−3 | 65.24 |
| | $J^{1-\text{Lip}_1}$ & tanh | 26.55 | 9.9e−2 | 50.42 | 2.7e−3 | 59.6 |
| | $J^2$ | 25.64 | 8.6e−2 | 53.85 | 2.5e−3 | 62.76 |
| | $J^{2-\text{Lip}_2}$ | 25.60 | 9.7e−2 | 50.71 | 2.7e−3 | 59.50 |
| | $J^2$ & tanh | 26.27 | 8.2e−2 | 54.81 | 2.5e−3 | 63.10 |
| | $J^{2-\text{Lip}_2}$ & tanh | 26.05 | 8.7e−2 | 53.24 | 2.5e−3 | 61.50 |
| ResNet34 | $J^0$ | **27.42** | 2.6e−2 | 90.41 | 0.07e−3 | 94.86 |
| | $J^{0-\text{Lip}_1}$ | 29.44 | 6.8e−2 | 62.38 | 1.7e−3 | 74.52 |
| | $J^{0-\text{Lip}_2}$ | 28.18 | 4.8e−2 | 70.94 | 1.4e−3 | 81.88 |
| | $J^0$ & tanh | 40.72 | 1.5e−2 | 81.19 | 0.05e−3 | 82.63 |
| | $J^{0-\text{Lip}_1}$ & tanh | 29.87 | 6.9e−2 | 60.83 | 1.7e−3 | 71.11 |
| | $J^{0-\text{Lip}_2}$ & tanh | 38.61 | 3.6e−2 | 68.34 | 1.0e−3 | 74.60 |
| | $J^1$ | 28.81 | 5.8e−2 | 66.59 | 1.4e−3 | 78.09 |
| | $J^{1-\text{Lip}_1}$ | 29.51 | 7.9e−2 | 56.73 | 2.0e−3 | 68.03 |
| | $J^1$ & tanh | 29.38 | 5.9e−2 | 62.45 | 1.6e−3 | 71.31 |
| | $J^{1-\text{Lip}_1}$ & tanh | 30.25 | **7.9e−2** | **55.6** | 2.1e−3 | **64.61** |
| | $J^2$ | 28.21 | 5.6e−2 | 66.12 | 1.5e−3 | 75.84 |
| | $J^{2-\text{Lip}_2}$ | 28.21 | 6.9e−2 | 66.12 | 1.8e−3 | 70.15 |
| | $J^2$ & tanh | 29.19 | 5.6e−2 | 64.51 | 1.7e−3 | 71.37 |
| | $J^{2-\text{Lip}_2}$ & tanh | 28.01 | 7.1e−2 | 58.40 | **2.1e−3** | 66.08 |
| ResNeXt34 (4x32d) | $J^0$ | 21.24 | 4.7e−2 | 74.18 | 1.4e−3 | 81.52 |
| | $J^{0-\text{Lip}_1}$ | 23.62 | 9.9e−2 | 51.05 | 2.6e−3 | 62.45 |
| | $J^{0-\text{Lip}_2}$ | 21.97 | 10.8e−2 | 47.64 | 3.2e−3 | 55.51 |
| | $J^0$ & tanh | 21.05 | 9.4e−2 | 52.28 | 1.3e−3 | 74.88 |
| | $J^{0-\text{Lip}_1}$ & tanh | 23.39 | 11.7e−2 | 46.99 | 3.2e−3 | 56.12 |
| | $J^{0-\text{Lip}_2}$ & tanh | 21.05 | 9.4e−2 | 52.28 | 2.7e−3 | 60.90 |
| | $J^1$ | 22.06 | 8.8e−2 | 53.09 | 2.5e−3 | 63.71 |
| | $J^{1-\text{Lip}_1}$ | 23.50 | 11.8e−2 | 47.22 | 3.1e−5 | 56.23 |
| | $J^1$ & tanh | 22.88 | 8.7e−2 | 53.71 | 2.5e−3 | 62.30 |
| | $J^{1-\text{Lip}_1}$ & tanh | 23.23 | **13.7e−2** | 43.59 | 3.7e−3 | 51.48 |
| | $J^2$ | 21.57 | 8.6e−2 | 53.77 | 2.6e−3 | 61.72 |
| | $J^{2-\text{Lip}_2}$ | 21.73 | 11.2e−2 | 46.79 | 3.2e−3 | 55.92 |
| | $J^2$ & tanh | **21.01** | 9.9e−2 | 50.33 | 3.1e−3 | 55.87 |
| | $J^{2-\text{Lip}_2}$ & tanh | 21.47 | 13.6e−2 | **42.58** | **4.0e−3** | **48.98** |

Table 6: CIFAR-100 stability statistics. Each column reports the maximum observed norm over the test data.

| Model | defense method | $\|\nabla\ell\|_2$ | $\|\nabla f\|_{2,\infty}$ | $\|\nabla\ell\|_1$ | $\|\nabla f\|_{1,1}$ |
|---|---|---|---|---|---|
| AllCNN | $J^0$ | 27.52 | 0.45 | 971.82 | 0.15 |
| | $J^{0-\mathrm{Lip}_1}$ | 15.20 | 0.29 | 539.26 | 0.09 |
| | $J^{0-\mathrm{Lip}_2}$ | 20.36 | 0.32 | 744.05 | 0.10 |
| | $J^0$ & tanh | 8.94 | 0.12 | 344.42 | 0.05 |
| | $J^{0-\mathrm{Lip}_1}$ & tanh | 20.20 | 0.34 | 727.95 | 0.10 |
| | $J^{0-\mathrm{Lip}_2}$ & tanh | 6.42 | 0.08 | 207.62 | 0.03 |
| | $J^1$ | 17.78 | 0.37 | 614.76 | 0.11 |
| | $J^{1-\mathrm{Lip}_1}$ | 15.67 | 0.26 | 561.61 | 0.08 |
| | $J^1$ & tanh | 23.88 | 0.40 | 846.45 | 0.13 |
| | $J^{1-\mathrm{Lip}_1}$ & tanh | 15.71 | 0.26 | 503.44 | 0.09 |
| | $J^2$ | 19.94 | 0.34 | 758.63 | 0.11 |
| | $J^{2-\mathrm{Lip}_2}$ | 16.22 | 0.27 | 569.32 | 0.08 |
| | $J^2$ & tanh | 23.18 | 0.38 | 809.10 | 0.11 |
| | $J^{2-\mathrm{Lip}_2}$ & tanh | 19.43 | 0.31 | 710.88 | 0.10 |
| ResNet34 | $J^0$ | 88.75 | 1.59 | 3235.35 | 0.76 |
| | $J^{0-\mathrm{Lip}_1}$ | 16.73 | 0.31 | 608.20 | 0.11 |
| | $J^{0-\mathrm{Lip}_2}$ | 27.61 | 0.47 | 1036.65 | 0.16 |
| | $J^0$ & tanh | 27.11 | 0.38 | 951.95 | 0.19 |
| | $J^{0-\mathrm{Lip}_1}$ & tanh | 15.74 | 0.30 | 591.40 | 0.09 |
| | $J^{0-\mathrm{Lip}_2}$ & tanh | 7.87 | 0.09 | 301.96 | 0.03 |
| | $J^1$ | 25.99 | 0.51 | 994.92 | 0.19 |
| | $J^{1-\mathrm{Lip}_1}$ | 15.19 | 0.28 | 534.66 | 0.11 |
| | $J^1$ & tanh | 24.02 | 0.41 | 811.88 | 0.18 |
| | $J^{1-\mathrm{Lip}_1}$ & tanh | 13.38 | 0.24 | 511.13 | 0.08 |
| | $J^2$ | 31.09 | 0.45 | 1054.41 | 0.33 |
| | $J^{2-\mathrm{Lip}_2}$ | 18.04 | 0.35 | 669.75 | 0.11 |
| | $J^2$ & tanh | 27.84 | 0.44 | 964.46 | 0.14 |
| | $J^{2-\mathrm{Lip}_2}$ & tanh | 16.81 | 0.27 | 639.93 | 0.08 |
| ResNeXt34 (4x32d) | $J^0$ | 93.83 | 1.89 | 3130.45 | 0.69 |
| | $J^{0-\mathrm{Lip}_1}$ | 22.53 | 0.36 | 774.36 | 0.18 |
| | $J^{0-\mathrm{Lip}_2}$ | 25.46 | 0.43 | 882.66 | 0.16 |
| | $J^0$ & tanh | 50.69 | 0.64 | 1401.32 | 0.29 |
| | $J^{0-\mathrm{Lip}_1}$ & tanh | 19.67 | 0.40 | 712.80 | 0.15 |
| | $J^{0-\mathrm{Lip}_2}$ & tanh | 9.91 | 0.12 | 352.78 | 0.04 |
| | $J^1$ | 34.60 | 0.71 | 1240.98 | 0.23 |
| | $J^{1-\mathrm{Lip}_1}$ | 21.66 | 0.36 | 816.33 | 0.14 |
| | $J^1$ & tanh | 41.69 | 0.74 | 1591.90 | 0.27 |
| | $J^{1-\mathrm{Lip}_1}$ & tanh | 15.81 | 0.28 | 564.42 | 0.10 |
| | $J^2$ | 44.81 | 0.73 | 1683.28 | 0.27 |
| | $J^{2-\mathrm{Lip}_2}$ | 27.58 | 0.46 | 1033.29 | 0.15 |
| | $J^2$ & tanh | 40.04 | 0.72 | 1398.56 | 0.23 |
| | $J^{2-\mathrm{Lip}_2}$ & tanh | 17.97 | 0.35 | 642.79 | 0.13 |

