# OpenReview forum: "Improved robustness to adversarial examples using Lipschitz regularization of the loss"
_ICLR.cc/2019/Conference_

### Official Review · AnonReviewer2 · 2018-11-01
**Interesting idea but poorly written**

**Rating:** 4
**Confidence:** 3

**Review:**

This paper explores augmenting the training loss with an additional gradient regularization term to improve the robustness of models against adversarial examples. The authors show that this training loss can be interpreted as a form of adversarial training against optimal L2 and L_infinity adversarial perturbations. This augmented training effectively reduces the Lipschitz constant of the network, leading to improved robustness against a wide variety of attack algorithms.

While I believe the results are correct and possibly significant, the paper is poorly written (especially for a 10 page submission) and comparison with prior work on reducing the Lipschitz constant of the network is lacking. The authors also made little to no effort in writing to ensure the clarity of their paper. I would like to see a completely reworked draft before opening to the idea of recommending acceptance.

Pros:
- Theoretically intuitive method for improving the model's robustness.
- Evaluation against a wide variety of attacks.
- Empirically demonstrated improvement over traditional adversarial training.

Cons:
- Lack of comparison to prior work. The authors are aware of numerous techniques for controlling the Lipschitz constant of the network for improved robustness, but did not compare to them at all.
- Poorly written. The paper contains multiple missing figure references, has a duplicated table (Tables 1 and 3), and the method is not explained well. I am confused as to how the 2-Lip loss is minimized. Also, the paper organization seems very chaotic and incoherent, e.g., the introduction section contains many technical details that would better belong in related works or methods sections.

--------------------------------------------

Revision:

I thank the authors for incorporating my suggestions and reworking the draft, and I have updated my rating in response to the revision. While I believe the organization is much cleaner and easier to follow, there is still much room for improvement. In particular, the paper does not introduce concepts in a logical order for a non-expert to follow (e.g. Reviewer 1) and leaps into the paper's core idea too quickly. I am strongly in favor of exceeding the suggested page limit of 8 pages and using that space to address these concerns.

A more pressing concern is the evaluation of prior work. The authors added a short section (Section 5.4) comparing their method to that of (Qian and Wegman, 2018). This is certainly a reasonable comparison and the results seem promising, the evaluation lacks an important dimension -- varying the value of epsilon and observing the change in robustness. This is an important aspect for defenses against adversarial examples as certain defense may be less robust but are insensitive to the adversary's strength. Showing the robustness across different adversary strengths gives a more informative view of the authors' proposed method in comparison to others. The evaluation is also lacking in breadth, ignoring other similar defenses such as (Cisse et al., 2017) and (Gouk et al., 2018).

---

> ### Author Response · Authors · 2018-11-24
> **Reply to AnonReviewer2**
>
> Thank you for your comments. Following your suggestion, we have completely reworked our submission, with an eye towards clarity and the page limit.  In fact we have reduced the page count to just over seven pages.  We hope that you find the paper better organized and easier to read.
>
> Regarding comparison to other results, we have re-analyzed our experimental results and have now included a direct comparison with state-of-the-art results. We find that when attacks are measured in the 2-norm, our method is state-of-the-art, improving on the previous state-of-the-art by 11%. When measured in the max-norm, our results are comparable to the state-of-the-art (Madry et al (2017)), however we use only one-step adversarial training, whereas in Madry et al seven step adversarial training is used.
>
> We have also now included a section explicitly comparing our methods with prior methods, which can be summarized as follows. Prior work has focused on controlling the estimate of the Lipschitz constant using the product of norms of weight matrices. We argue that for deep networks this estimate is inaccurate, since its error grows exponentially in the number of layers. In our work we propose an alternative method for estimating the Lipschitz constant, which is an underestimate, and is estimated from the training data. This is a novel approach.
>
> Please also see the general reply to all reviewers, above.

---

### Official Review · AnonReviewer3 · 2018-11-02
**Interesting idea -- could be significantly strengthened**

**Rating:** 6
**Confidence:** 3

**Review:**

Summary: this paper uses a trick to simplify the adversarial loss by one in which the adversarial perturbation appears in closed form.

pros:

- interesting idea
- experiments are interesting

cons:

- formal results are either trivial or could be improved in their statements
- experimental guarantees only, up to what is hidden in the Big-Oh notations of Theorem 2.2, 2.3.

details:

* In Theorem 2.2, you need to remove the $O(epsilon^2)$, unless you point to the Taylor theorem that guarantees that for the identity you claim before (5). The closest one I see is that the O(||a||^2) is in fact $||a|| u(||a||)$ with $\lim u(x) = 0$ as $x \rightarrow 0$, which does not guarantee the $O$ notation for any $a$.

* In Theorem 2.2, how do you pass from the solution of (5) (which is indeed a vector) to the solution of the following equation, which, without constraint, gives a dim > 1 subspace in the general case ?

* In all cases, you do not get Theorem 2.3 in its form as the $O$ notation just guarantees you an upperbound. You need to rephrase.

* Figure ?? (twice) before Section 3

* Define the “group norm” notation appearing with the max in (8) (isn’t one redundant ?)

* Section 3.4 is interesting. Have you looked at generalising your observation in the last identity to more losses  = f-divergences (hence, proper losses modulo assumptions) ?

* Section 4: many Figure ??

---

> ### Author Response · Authors · 2018-11-24
> **Reply to AnonReviewer3**
>
> Thank you for your review. We have reworked our draft, and we hope that our new version addresses your points.
>
> We would like to make a comment regarding Big-O notation. In the context used in the paper, which corresponds to https://en.wikipedia.org/wiki/Big_O_notation ,  Big-O notation is a rigorous result, not experimental.
>
> In many areas of scientific computing, engineering and statistics it is accepted that results need only be shown up to Big-O of some error term. For example in polynomial interpolation it typically it suffices to show a particular method has error epsilon^(n+1) with a n-th degree polynomial. We have shown that adversarial training is equivalent to Total Variation minimization, up to order epsilon^2, where epsilon is the size of the adversarial perturbation. This means that when epsilon is small, as is typical (our epsilon is 0.01),  the two methods are nearly equivalent.  By equivalent,  lossely speaking, we mean that replacing one term with another should lead to results which are very close.   However the Big-O notation has a rigorous meaning in the limit as \epsilon goes to zero.
>
> Please also see our general reply to all reviewers, above.

---

### Official Review · AnonReviewer1 · 2018-11-05
**Possibly a good paper but not my area of expertise at all**

**Rating:** 6
**Confidence:** 1

**Review:**

The authors propose a novel method of training neural networks for robustness of adversarial attacks based on 2-norm and Lipschitz regularization. Unfortunately I'm not at all familiar with the literature on adversarial attacks so it is difficult for me to judge the quality and significance of this work. The theoretical results look plausible and clearly stated. The experiments show improvements over existing methods but I can't tell whether the right baselines were used. Overall the writing is reasonably clear but not very accessible for someone not already familiar with the area.

---

> ### Author Response · Authors · 2018-11-24
> **Reply to AnonReviewer1**
>
> We have posted a new version of our paper. We have re-written the paper to be as accessible as possible to someone not directly familiar with adversarial robustness. We have also pushed the heavier math to the appendix, and reinterpreted our Lipschitz regularization as worst-case adversarial training, which is an interpretation of a more familiar idea in the area.
>
> We would greatly appreciate your comments on the new draft. We hope that you will find it easier to read.
>
> Please also see our general reply to all reviewers above.

---

### Author Response · Authors · 2018-10-18
**Results measured in L-infinity**

Several people have suggested that it would be helpful if we also reported measurements of adversarial distance in the L-infinity norm (to complement L2). Following this suggestion, we have re-generated all the tables and figures in L-infinity.

For example, our main results are presented in Table 1, where we report median adversarial distance and percent error at a fixed adversarial distance. Here is Table 1 with distances in L-infinity. We report percent misclassified at adversiarial distance 1/16 (rather than 0.1) to more easily compare with other literature's results.

Dataset     defense method     median distance    % err at eps=1/16

CIFAR-10    J0 (baseline)                            1.02e-2                          99.92
                    J1 (AT, FGSM)                          2.12e-2                          96.06
                    J2 (AT, L2)                                3.45e-2                          84.76
                   J2-Lip & tanh                            6.00e-2                          51.64

CIFAR-100   J0 (baseline)                           5.83e-3                          99.61
                     J1 (AT, FGSM)                         1.07e-2                          98.46
                     J2 (AT, L2)                               1.06e-2                           98.03
                    J2-Lip & tanh                           1.60e-2                           93.73

We hope this updated table is useful during the review process.

---

### Public Comment · ~Oleg_Trott1 · 2018-11-13
**Lemma 3.3 is incorrect**

I don't think Lemma 3.3 is correct. As I understood it, the Lemma claims that to calculate a particular Lipschitz constant (2,inf) of a feed-forward network with entry-wise 1-Lipschitz nonlinearities, one can ignore the nonlinearities (and of course the biases).

Please consider this runnable Numpy code as a counterexample. The network is defined by f. The product of the matrices w1 and w0 is 0. However, the network generates distinct outputs f(x1) and f(x2):


import numpy as np
from numpy.linalg import norm

def relu(x): return np.maximum(x, 0)

w0 = np.array([[1., -1.], [-1., 1.]])
w1 = np.array([[1., 1.]])

def f(x): return w1.dot(relu(w0.dot(x)))

def lip_lower_bound(x1, x2): return norm(f(x1) - f(x2), np.inf) / norm(x1 - x2, 2)

x1 = np.array([0., 0.])
x2 = np.array([-1., 1.])

print(w1.dot(w0)) # 0

print(lip_lower_bound(x1, x2)) # sqrt(2)

---

> ### Author Response · Authors · 2018-11-22
> **The lemma was not needed, it was redundant, so it has been removed**
>
> Hi,
> Thanks for your comment.  We had in mind a result which would be true under additional assumptions.  But, in fact, we don't use the lemma anywhere - it was just to illustrate upper bounds on the Lipschitz constant coming from the architecture.
>
> We are removing it from the revision.

---

### Author Response · Authors · 2018-11-24
**General reply to reviewers**

Thank you to the reviewers for your comments. We have posted a new draft of the manuscript. Following AnonReviewer2’s comments, we have completely reworked the draft. We have attempted to communicate our ideas and results as clearly as possible. We hope that the paper is accessible to persons outside the field of adversarial attacks as well, such as AnonReviewer1.

We would like to highlight the merits of our results.

1. We achieve state-of-the-art results when measuring attacks in the 2-norm, improving by 11% over the previous state-of-the-art on CIFAR-10. Following off line conversations with some of our colleagues, we have also analyzed our results in the max-norm as well. In the max-norm, we are on par with the current state-of-the-art (Madry et al (2017)). However we did not focus our efforts on the max-norm, so we believe with a bit more effort our results could possibly be improved, for example if we had used multi-step adversarial training like in Madry et al (we only used one-step adversarial training).

2. Our implementation of Lipschitz regularization is novel and an improvement to existing results in terms of both accuracy (by orders of magnitude) and efficiency (we can leverage the gradients already used in adversarial training). Training networks by penalizing with estimates of the Lipschitz constant have in the past used the product of weight matrix norms. However this estimate of the Lipschitz constant grows exponentially in the number of layers. As such the estimate is intractable for deep networks. In contrast, our method provides a more accurate estimate, which we demonstrate is closer in magnitude to the true value of the Lipschitz constant.

3. We believe that our interpretation of adversarial training will be interesting and useful to the community. We show that adversarial training is a form of Total Variation regularization, which has been used successfully outside the deep learning community in image preprocessing to denoise images. We believe that this is a useful insight that could be leveraged further in the future.

4. We obtain novel average case and worst case robustness bounds, which we verify empirically.  These bounds allow use to predict adversarial robustness based on the statistics of quantities we can read off of the trained model.

---

### Public Comment · (anonymous) · 2018-11-29
**pointer to discussion**

There is a thread discussing this paper, which reviewers may be interested in at:
https://openreview.net/forum?id=ByxGSsR9FQ&noteId=Bklz239KCX

---

> ### Author Response · Authors · 2018-11-29
> **thanks for the pointer**
>
> I see there is a discussion of how our results compare.  Some interesting points are raised by the poster and by the authors of the other paper.
>
> We stand by our statement in our paper:
> We implemented the attacks correctly and our robustness results are stronger than the other paper, and any other published results for L2 attacks.

---

### Public Comment · (anonymous) · 2018-12-05
**about training details**

This paper looks quite interesting to connect Lipschitz regularization with min-max adversarial training. If my understanding is correct, the training process will be performed on (1). Since the norm of gradient with respect to x is penalized, solving (1) requires the gradient of \partial | \nabla_x l(x) |  / \partial \theta, where \theta are network parameters. Is such quantity easy to compute? Compared to min-max adversarial training, is there a significant speed-up using Lipschitz regularization?

---

> ### Author Response · Authors · 2018-12-06
> **training details**
>
> Hi,
>
> Thanks for your interest. You're correct, a mixed derivative -- in both x (image) and theta (parameters) is computed. Our implementation was easily done in PyTorch, simply by running autograd twice - first in x to get the norm gradient, then in theta. In practice we found that networks trained with this Lipschitz penalty took no longer than four times the training time of an unregularized network. Which means that if you are doing many steps of PGD adversarial training (for example in Madry et al uses 7 steps), Lipschitz regularization can be faster. The comparison depends on how many PGD steps you take.

---

### Meta-Review · Area_Chair1 · 2018-12-16
**The quality of the presentation makes it hard to properly assess the quality of the results**

**Confidence:** 3
**Recommendation:** Reject

**Metareview:**

This paper suggests augmenting adversarial training with a Lipschitz regularization of the loss, and suggests that this improves the adversarial robustness of deep neural networks. The idea of using such regularization seems novel. However, several reviewers were seriously concerned with the quality of the writing. In particular, the paper contains claims that not only are not needed but also are incorrect. Also, the Reviewer 2 in particular was also concerned with the presentation of prior work on Lipschitz regularization.

Such poor quality of the presentation makes it impossible to properly evaluate the actual paper contribution.